# Presepsin cut-off value for diagnosis of sepsis in patients with renal dysfunction

**Kimika Arakawa**⬥\*, **Ayako Saeki**◉, **Reo Ide**◉, **Yoshiteru Matsushita**

Department of Clinical Laboratory, Clinical Research Institute, National Hospital Organization Kyushu Medical Center, Fukuoka, Japan

◉ These authors contributed equally to this work.
\* arakawa.kimika.rp@mail.hosp.go.jp

**Data Availability Statement:** All relevant data are within the paper and its Supporting Information files.

**Funding:** The authors received no specific funding for this work.

## Abstract

Presepsin is used as a marker for diagnosing sepsis, but its serum concentration is affected by renal function. We investigated the effect of the estimated glomerular filtration rate (eGFR) determined by creatinine on the diagnostic accuracy of presepsin to identify the optimal cut-off value in patients with renal dysfunction. A total of 834 patients aged ≥18 years with serum presepsin and creatinine measured on the same day over a period of 1 year were included. Sepsis was diagnosed in three ways: sepsis-1, sepsis-3, and clinical diagnosis (Sep-C). Pre-sepsin showed a significant negative correlation with eGFR (r = −0.55, p<0.01), with median and interquartile ranges of presepsin values for patients in each eGFR category as follows: ≥90, 263 (169–460); ≥60–<90, 309 (205–578); ≥45–<60, 406 (279–683); ≥30–<45, 605 (379–1109); ≥15–<30, 1027 (675–1953); <15, 1977 (1199–3477); and on hemodialysis, 3964 (2343–6967). In receiver operating characteristic (ROC) analysis, the area under the curve (AUC) for sepsis-1 was the lowest (0.64 ± 0.02), while Sep-C (0.80± 0.03) and sepsis-3 (0.75 ± 0.03) were moderately accurate. Comparing AUCs after dividing patients into eGFR ≥60 and <60 showed that the AUC of Sep-C was lower in the eGFR ≥60 group, while the AUC of sepsis-3 was ≥ 0.7 in both groups. The following cut-offs were obtained by ROC analysis for sepsis-3: 466 pg/mL in the ≥60 group and 960 pg/mLin the < 60 group. Presep-sin facilitated diagnosis sepsis based on sepsis-3 criteria regardless of renal function. We found that the optimal cut-offs for patients in this study were 500 pg/mL for eGFR ≥ 60 and 1000 pg/mL for < 60. However, future prospective diagnostic studies on sepsis-3 are needed to determine the cut-offs for patients with renal dysfunction.

## Introduction

Sepsis-1 (Sep-1) was introduced in 1992 to aid the definition and diagnosis of sepsis [1]. This was revised in 2003 to sepsis-2 [2], and sepsis-3 (Sep-3) was further proposed as a new diagnostic criterion for sepsis in 2016 [3]. The first edition of the Japanese version of the Sepsis Practice Guidelines was published in 2012, followed by the revised edition in 2016 [4]. These guidelines list procalcitonin, presepsin (Pre), and interleukin-6 as biomarkers of sepsis. Of these, Pre, as a subtype of soluble CD14, was a novel sepsis marker discovered in Japan in 2004

**Competing interests:** The authors have declared that no competing interests exist.

[5]. Unlike the previously used marker procalcitonin, Pre has been reported to be advantageous for detecting sepsis in patients with severe trauma and burns [6, 7]. In addition, Pre has been reported to have a short response time and to be increased in the early stage of infection [8, 9], and its levels strongly reflect the patient's clinical course [9, 10]. However, Pre is excreted from the kidney and its concentration thus correlates with renal function; its levels were shown to be high in patients with renal dysfunction [11, 12]. Following a request from the emergency department, the in-hospital sepsis marker in our hospital was changed from procalcitonin to Pre on 1 August, 2017. This study therefore aimed to confirm the relationship between renal function and Pre in clinical practice, and to investigate the effect of renal dysfunction on the diagnostic value of Pre. Furthermore, we attempted to determine the optimal cut-off value of Pre according to the degree of renal dysfunction.

## Materials and methods

### Design and subjects

This was a retrospective single-center study. First, we extracted the data of patients who had their Pre and creatinine (Cr) levels measured on the same day in our hospital between 1 August, 2017, and 31 July, 2018. It is presumed that these patients had symptoms of infectious disease (e.g., fever) and needed to be tested for sepsis. For patients with multiple measurements taken throughout the year, we used the initial value. We included inpatients or outpatients over the age of 18 years. We excluded patients without at least two measurements of body temperature, heart rate, respiratory rate, and white blood cell counts. It is expected that some of these patients had low muscle mass and overestimated values of the estimated glomerular filtration rate (eGFR). We excluded seven patients with extremely high eGFRs ($\geq$200 mL/min/ 1.73 m$^2$) and one pregnant woman. As the top 2.5% of the eGFR distribution in all patients was 139.4 pg/mL, we investigated 13 patients with $\geq$140 pg/mL and excluded nine patients with a body mass index of <20 kg/m$^2$.

This study was conducted in accordance with our institutional guidelines and approved by the Ethics Committee of the National Hospital Organization, Kyushu Medical Center (18C009). This study analyzed medical information acquired during usual medical care. This study was a retrospective, non-invasive, and non-intervening study, and thus we did not obtain written or verbal informed consent. Our Institutional Review Board waived the requirement for informed consent. Nevertheless, we have posted a notice on our hospital homepage that provides information regarding this study to eligible patients and their families and allows them to opt-out. No patients denied participation in this research. In addition, we anonymized the data that were analyzed and stored.

### Data collection and definition of diagnosis

We collected information on blood pressure, pulse rate, complete blood count, blood chemistry, and blood gases on the same day as the Pre measurement from the patients' medical records. In addition, we retrospectively investigated diagnoses of infections and the sites of infections based on medical examinations and blood and imaging tests from their medical records. A diagnosis of sepsis was made by three methods: Sep-1: infectious patients with at least two systemic inflammatory response syndrome (SIRS) criteria (American College of Chest Physicians/Society of Critical Care Medicine Consensus Conference Committee, 1992 [1]); Sep-3: infectious patients diagnosed according to Sepsis-3 [3]; and Sep-C: infectious patients diagnosed with sepsis in their medical records. Sep-C signified that we retrospectively investigated the medical record in which the doctor had listed the diagnosis of sepsis.

## Estimation of renal function

Serum Cr concentrations were measured by the enzymatic method using an Aqua-auto Kainos CRE-III plus Test Kit (Kainos, Tokyo, Japan). The eGFR according to Cr level in each participant was calculated using the equation provided by the Japanese Society of Nephrology as follows: eGFR (mL/min/1.73 m$^2$) = 194 × Cr (mg/dl)$^{-1.094}$ × age$^{-0.287}$ (if female, × 0.739). GFR was categorized according to the KDIGO 2012 by eGFR [13]. eGFR categories were defined as follows: ≥90, ≥60–<90, ≥45–<60, ≥30–<45, ≥15–<30, and <15 mL/min/1.73 m$^2$. Patients on maintenance hemodialysis (HD) were also evaluated.

## Measurement of Pre

Serum samples were collected into 8 mL vacuum-sealed blood collection tubes for Pre measurement. Serum Pre concentrations were measured using a STACIA CLEIA Pre assay kit (LSI Medience Corporation, Tokyo, Japan) based on a chemiluminescent enzyme immunoassay. The measurement range of the assay was 50–20,000 pg/mL.

## Statistical analysis

All values, except for Pre, are shown as the median and interquartile range (IQR). Pre data were analyzed by a non-parametric test because of the skewed distribution. The correlation between Pre (log-transformed Pre) and eGFR was assessed by Spearman's analysis. Comparison of sepsis frequency by eGFR category was performed by logistic analysis adjusted by sex and age. Pre values were compared between the sepsis vs. non-sepsis groups as non-parametric variables using the Mann–Whitney U test and among multiple eGFR categories using Dunnett-Hsu's analysis adjusted by sex and age. The diagnostic ability of Pre was evaluated by receiver operating characteristic (ROC) curve analysis. The cut-off values were determined as the points where sensitivity and specificity were equal. A p value of <0.05 was considered significant. Statistical analyses were performed using SAS ver.9.4 (SAS Institute Japan Ltd., Tokyo, Japan). Stat Flex ver.6 (Artech Co., Ltd. Osaka, Japan) was used to create graphs.

# Results

## Subject characteristics and diagnosis of sepsis

A total of 834 subjects [mean ± SD of age: 69.9 ±15.8 years, 353 females (42.3%)] were included, of whom 594 (71.2%) had infections. Of overall, 282 patients (33.8%) were diagnosed with sepsis based on Sep-1, 107 patients (12.8%) based on Sep-3, and 71 patients (8.5%) based on medical records (Sep-C). A total of 115 (13.8%) patients with infections were not evaluated for Sep-3 because their SOFA and qSOFA scores could not be calculated. Our retrospective medical record survey revealed patients who could not be diagnosed with sepsis based on the qSOFA score because there were no records of respiratory rate or change of consciousness in non-ICU patients. In addition, some ICU patients could not be diagnosed with sepsis by SOFA score because there were no records of the partial pressure of oxygen in arterial blood (PaO$_2$)/fraction of inspiratory oxygen (FiO$_2$), platelet count, bilirubin level, or Glasgow Coma Scale. Table 1 shows the patient backgrounds of the overall and Pre ≥433 pg/mL or <433 pg/mL groups. Compared with the <433 group, the ≥433 group was significantly older (73.0 ± 12.6 vs. 66.8 ± 17.8 years, p<0.01), had fewer women (34.0% vs. 50.6%, p<0.01), had more infectious patients (80.6% vs. 62.1%, p<0.01), had more patients with sepsis according to each criterion (Sep-1: 45.2% vs. 22.3%, p<0.01; Sep-3: 21.8% vs. 3.8%, p<0.01; Sep-C: 15.4% vs. 1.7%, p<0.01), had poorer renal function (Cr: 1.12 vs. 0.71 mg/dl, p<0.01; eGFR:

**Table 1. Clinical characteristics of the patients (overall and divided by the median Pre value).**

| Characteristic¤ | All (N = 834)¤ | Pre≥433¤ | Pre<433¤ |
|---|---|---|---|
| Age, years | 69.9 ± 15.8 | 73.0 ± 12.6* | 66.8 ± 17.8 |
| Sex | M: 481 (57.7%); F: 353 (42.3%) | M: 275 (66.0%); F: 142 (34.0%)†† | M: 206 (49.4%); F: 211 (50.6%) |
| Infectious disease | 595 (71.3%) | 336 (80.6%)†† | 259 (62.1%) |
| Respiratory | 227 (38.2%)[a] | 129 (38.4%)[a] | 99 (38.1%)[a] |
| Urinary tract | 66 (11.1%)[a] | 37 (11.0%)[a] | 29 (11.6%)[a] |
| Abdominal | 57 (9.6%)[a] | 31 (9.2%)[a] | 27 (10.4%)[a] |
| Others (including unknown) | 245 (41.1%)[a] | 139 (41.4%)[a] | 104 (39.9%)[a] |
| Sepsis by sepsis-1 criteria | 282 (33.8%) | 189 (45.2%)†† | 93 (22.3%) |
| Sepsis by sepsis-3 criteria[b] | 107 (12.8%) | 91 (21.8%)†† | 16 (3.8%) |
| Sepsis by Sep-C | 71 (8.5%) | 64 (15.4%)†† | 7 (1.7%) |
| Patients in ICU | 108 (13.0%) | 80 (19.2%)†† | 28 (6.7%) |
| Serum creatinine [mg/dl, median (IQR)] | 0.82 (0.63–1.27) | 1.12 (0.76–2.26)** | 0.71 (0.57–0.88) |
| eGFR (mL/min/1.73 m$^2$) | 62.5 ± 31.3 | 48.4 ± 31.4** | 76.6 ± 24.0 |
| eGFR categories | | | |
| ≥90 | 167 (20.0%) | 46 (11.0%) | 121 (29.0%) |
| ≥60–<90 | 292 (35.0%) | 104 (24.9%) | 188 (45.1%) |
| ≥45–<60 | 129 (15.5%) | 62 (14.9%) | 67 (16.1%) |
| ≥30–<45 | 94 (11.3%) | 61 (14.6%) | 33 (7.9%) |
| ≥15–<30 | 75 (9.0%) | 68 (16.3%) | 7 (1.7%) |
| <15 | 45 (5.4%) | 44 (10.6%) | 1 (0.2%) |
| HD | 32 (3.8%) | 32 (7.7%) | 0 (0.0%) |
| Serum presepsin [pg/mL, median (IQR)] | 433 (236–977) | 977 (617–1921)** | 236 (170–318) |
| Positive blood culture cases[c] | 97 (11.6%) | 71 (17.0%) | 26 (6.2%) |

Values are provided as the mean ± SD or n (%), unless otherwise specified.

M: male; F: female; Sep-C: sepsis by medical records; ICU: intensive care unit; eGFR: estimated glomerular filtration rate; HD: patients on maintenance hemodialysis; IQR: interquartile range.

**p<0.01 vs. Pre <433 using the Mann–Whitney U test.

††p<0.01 vs. Pre <433 by the chi-square test.

[a]Percentages of infection locations in patients with infectious diseases (n = 595).

[b]115 patients (13.8%) not evaluated.

[c]346 patients (41.5%) not examined.

48.4 ± 31.4 vs. 76.6 ± 24.0 mL/min/1.73 m$^2$, p<0.01), and had more intensive care unit patients (19.2% vs. 6.7%, p<0.01).

## Relationship between serum Pre and renal function

Pre and log-transformed Pre showed a significant negative correlation with eGFR (r = −0.55, p<0.01) (Fig 1). The distributions and median (IQR) values of Pre according to eGFR categories are shown in Fig 2. Compared with Pre in the ≥90 group, Pre was significantly higher in all groups except the ≥60–<90 group. The median Pre (IQR) values according to the eGFR categories were as follows: ≥90: 263 (169–460); ≥60–<90: 309 (205–578); ≥45–<60: 406 (279–683); ≥30–<45: 605 (379–1109); ≥15–<30: 1027 (675–1953); <15: 1977 (1199–3477); and HD: 3964 (2343–6967).

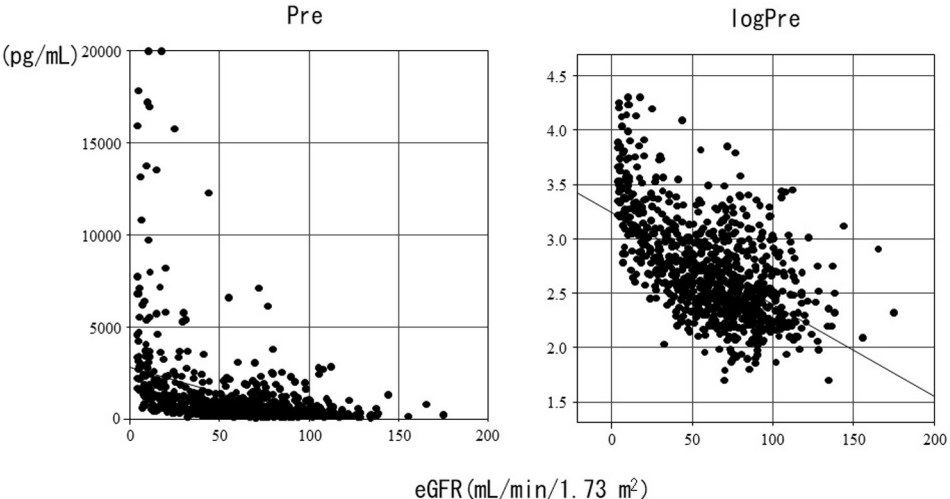

**Fig 1. Correlation between Pre and log-transformed Pre values and eGFRs in all subjects.** eGFR: estimated glomerular filtration rate; Pre: presepsin value. logPre: log-transformed presepsin value. Spearman's correlation coefficient = −0.55, p<0.01.

## Frequency of sepsis by eGFR categories

The frequency of Sep-1 was significantly higher in the ≥30–<45 group than that in the ≥90 group. However, Sep-C and Sep-3 were more frequent in patients with eGFRs <45 compared with those with eGFRs ≥90 (Fig 3).

## eGFR level and diagnostic accuracy of Pre for sepsis

On the basis of the above results, we classified the degree of renal function into groups with eGFRs ≥60 or <60, as approximately half of the patients with infection (n = 595) had eGFRs

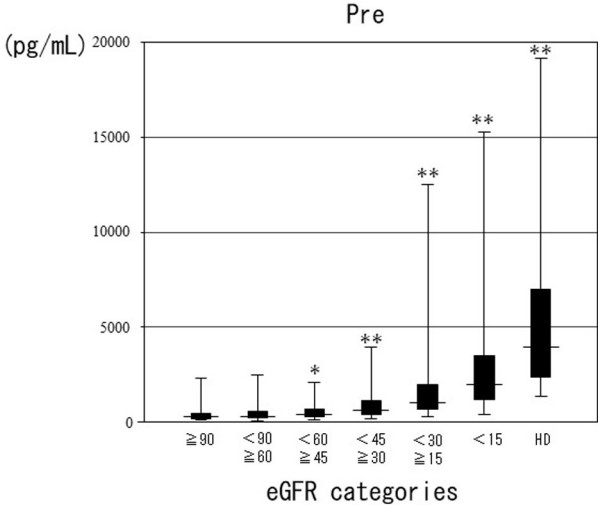

**Fig 2. Pre values in different eGFR categories.** eGFR: estimated glomerular filtration rate; Pre: presepsin value; ns: not significant. The horizontal bars represent the median, the boxes represent the IQR, and the vertical bars represent the upper and lower limits. p values calculated by Dunnett-Hsu's analysis adjusted by sex and age. *p<0.05 **p<0.01 compared with ≥90.

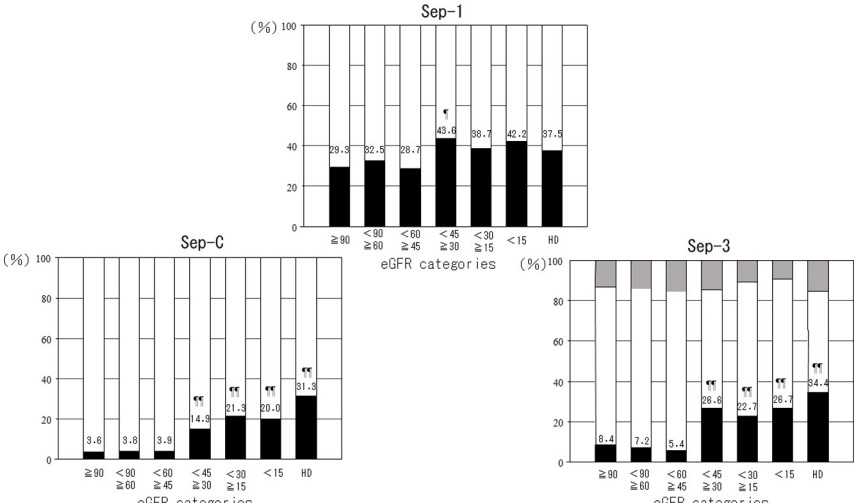

**Fig 3. Percentages of patients with sepsis according to eGFR categories.** Sep-1: sepsis diagnosed by Sep-1 criteria; Sep-C: sepsis diagnosed according to medical records; Sep-3: sepsis diagnosed by Sep-3 criteria. Black bars show patients with sepsis, white bars show patients without sepsis, and gray bars show subjects who were not evaluated because of lack of data. The numbers in the figure show percentages of patients with sepsis. p values calculated by logistic analysis adjusted by sex and age. ¶p<0.05, ¶¶p<0.01 compared with ≥90.

≥60 (N = 315, 52.9%) and the least difference in the frequency of sepsis between these groups (Sep-3: ≥60 11.1% vs. <60 25.7%, ≥45 10.4% vs. <45 34.0%, ≥30 14.0% vs. <30 34.5%; Sep-C: ≥60 5.4% vs. <60 19.2%, ≥45 5.4% vs. <45 25.7%, ≥30 7.5% vs. <30 30.2%). Pre differed significantly between patients with and without sepsis according to each diagnostic criteria and between patients with eGFRs ≥60 and <60. In summary, regardless of kidney function, Pre was significantly higher in patients with sepsis compared with that in patients without sepsis (Figs 4–6). We also investigated the accuracy of Pre for sepsis diagnosis based on each diagnostic criterion by ROC curve analysis in all subjects. The AUC for Sep-1 was the lowest (0.64 ± 0.02), and Sep-C (0.80 ± 0.03) and Sep-3 (0.75 ± 0.03) were moderately accurate. We performed similar analyses for eGFRs ≥60 and <60 and compared the AUCs. The AUC for Sep-1 was low in both groups (≥60: 0.67 ± 0.03 vs. <60: 0.62 ± 0.03), the AUC for Sep-C was higher in the eGFR<60 group (≥60: 0.68 ± 0.07 vs. <60: 0.78 ± 0.03), and the AUC for Sep-3 was almost equivalent in both groups (≥60:0.75 ± 0.04 vs. <60: 0.71 ± 0.04) (Figs 7–9). Finally, we determined the cut-offs for sepsis diagnosis from the ROC curves of Sep-3. According to this, the cut-off was 466 pg/mL (sensitivity = specificity 70%) in the ≥60 group and 960 pg/mL (sensitivity = specificity 67%) in the <60 group. From these results, we prioritized specificity and assumed an approximate optimal Pre cut-off for the diagnosis of sepsis in patients with renal dysfunction of 500 pg/mL for ≥60 and 1000 pg/mL for <60.

## Discussion

The current study found a significant negative correlation between eGFR and Pre. There was no significant difference in Pre between the groups of ≥90 and ≥60–<90, and in each of the other groups, Pre was significantly higher in the group with lower eGFR than in the group with ≥60. Similarly, the frequencies of sepsis diagnosed by Sep-C and Sep-3 were significantly higher in the group of <45 than ≥60. Numerous studies have reported a significant correlation between Pre and renal function in subjects with or without infection or sepsis [11, 12, 14–16]. The current results were consistent with these previous reports; however, few studies have

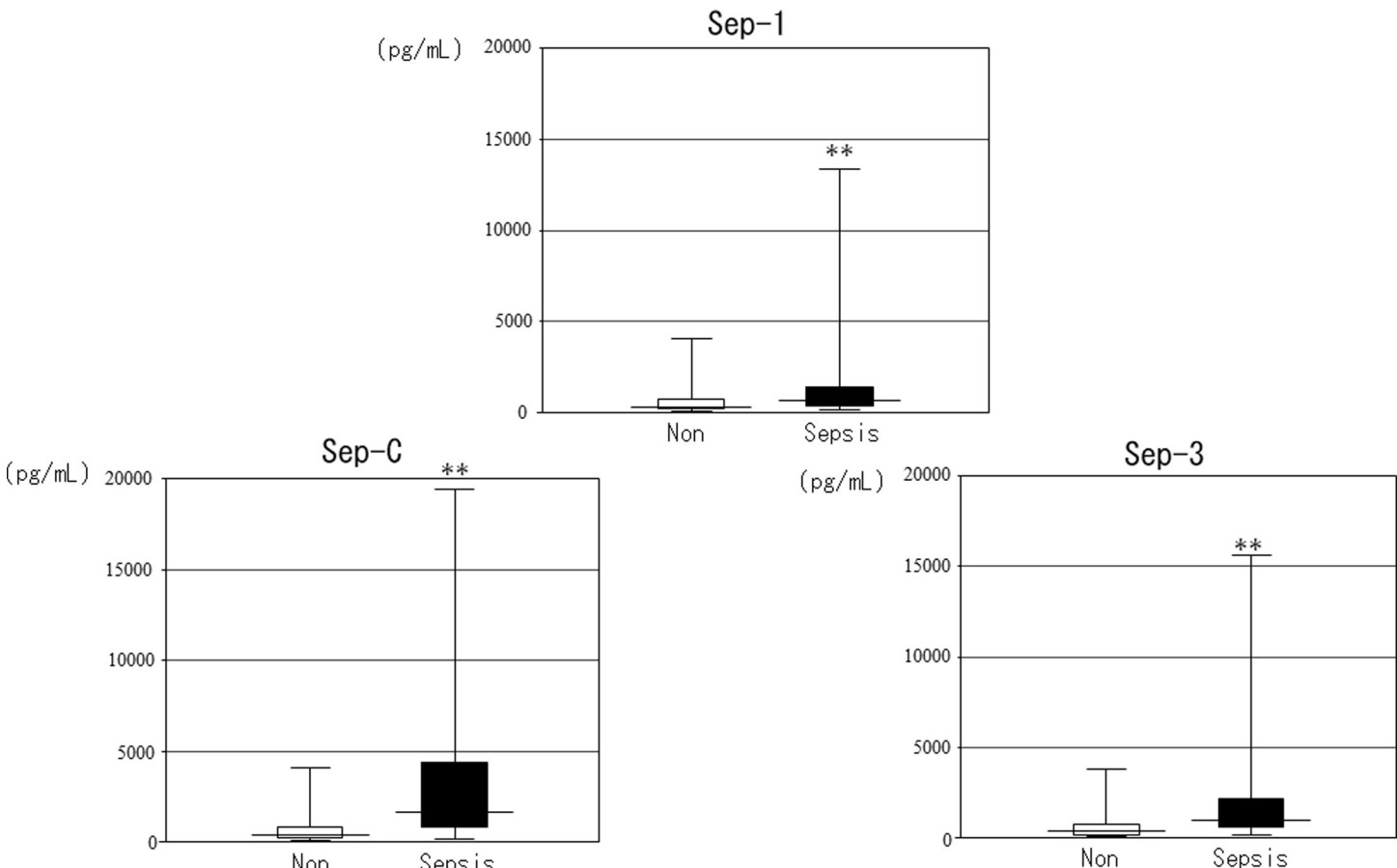

**Fig 4. Comparison of Pre levels in patients with and without each sepsis diagnosis [All subjects with infectious diseases (N = 595)].** The horizontal bars represent the median, the boxes represent the IQR, and the vertical bars represent the upper and lower limits. Non: The group of non-septic patients. Sepsis: The group of septic patients. The median Pre (IQR) values are as follows: Sep-1: Sepsis (N = 282), 693 (351–1400) vs. Non (N = 313), 392 (230–874) pg/mL; Sep-C: Sepsis (N = 72), 1628 (867–4372) vs. Non (N = 524), 454 (263–954) pg/mL; Sep-3: Sepsis (N = 107), 1036 (600-–2152) vs. Non (N = 373), 430 (117–874) pg/mL. *p<0.05, **p<0.01 by the Mann–Whitney U test vs. non-sepsis.

included patients with suspected infectious diseases and renal dysfunction, and the present results may thus be clinically meaningful. Moreover, no reports have compared Pre concentrations in relation to eGFR level, including patients with infections. Pre was significantly higher in patients with sepsis than in those without sepsis according to all three diagnostic criteria and among all subjects, or only <60 or ≥60 patients.

However, the diagnostic predictive ability of Pre differed depending on the type of diagnostic criteria. In the ROC analysis, the diagnostic accuracy of Pre was low for Sep-1 but moderate for Sep-3 and Sep-C, with AUCs >0.7. The effects of renal dysfunction on the predictive ability of Pre also differed according to the three criteria. That is, Sep-1 had a low AUC regardless of renal function, Sep-C had a high AUC in the <60 group, and Sep-3 had an AUC of ≥0.7 in both the ≥60 and <60 group. Sep-1 was proposed after repeated revisions based on the old diagnostic criteria for infectious diseases with SIRS. For this reason, we speculate that the diagnostic criteria for Sep-1 are questionable, resulting in inferior diagnostic accuracy of Pre. Sep-C was stipulated that there is a diagnosis of sepsis in the medical record or summary at the time of discharge. There are no clear criteria for this diagnosis, but Pre values are expected to play a role. In other words, it is possible that AUC was high in patients with eGFR<60 because

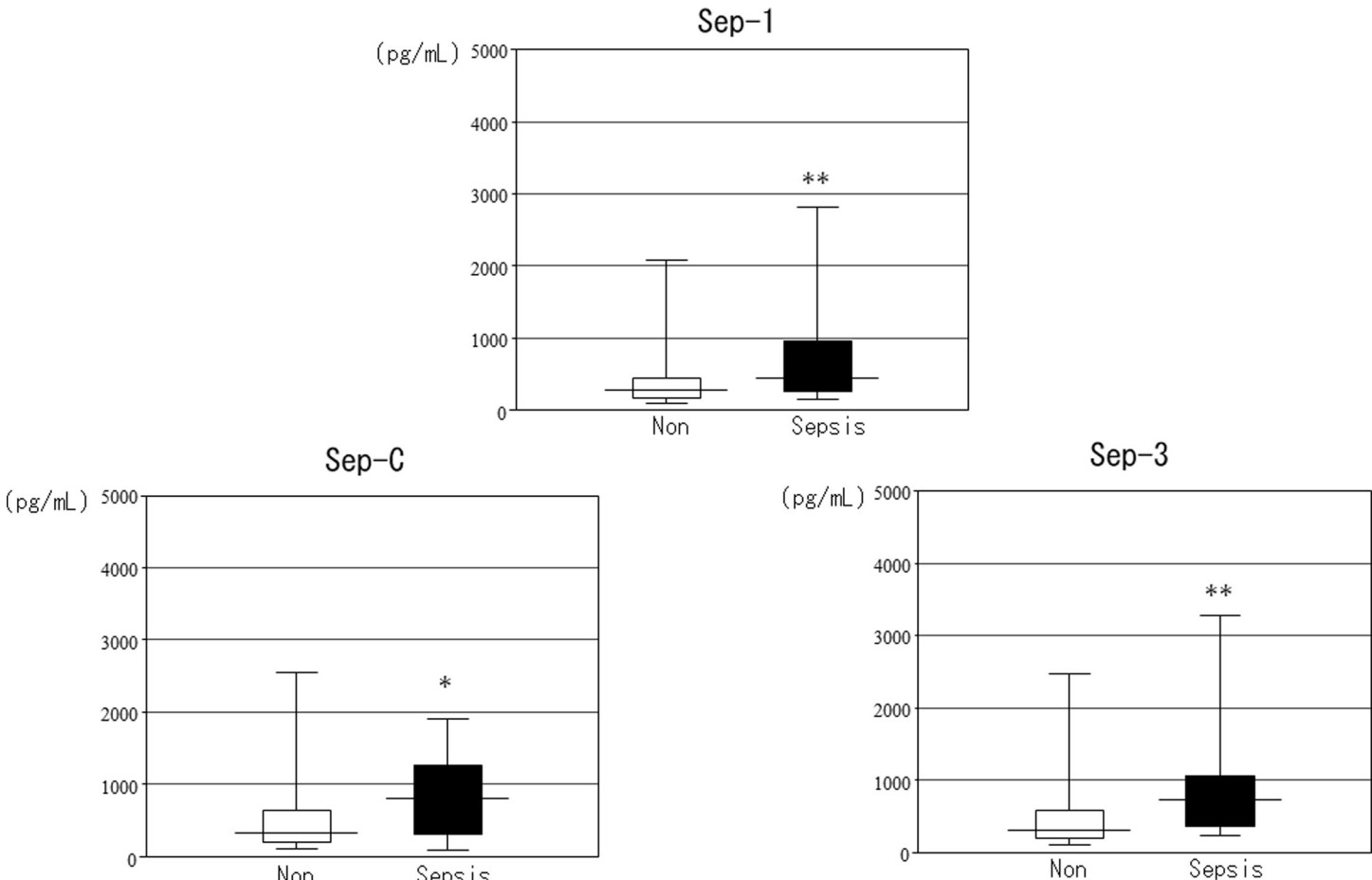

**Fig 5. Comparison of Pre levels in patients with and without each sepsis diagnosis[patients with eGFR≥60 (N = 315)].** The horizontal bars represent the median, the boxes represent the IQR, and the vertical bars represent the upper and lower limits. Non: The group of non-septic patients. Sepsis: The group of septic patients. The median Pre (IQR) values are as follows: Sep-1: Sepsis (N = 144), 445 (262–957) vs. Non (N = 171), 274 (176–448) pg/mL; Sep-C: Sepsis (N = 17), 802 (311–1267) vs. Non (N = 298), 322 (206–640) pg/mL; Sep-3: Sepsis (N = 35), 732 (367–1062) vs. Non (N = 216), 299 (195–588) pg/mL. *p<0.05, **p<0.01 by the Mann–Whitney U test vs. non-sepsis.

Pre was high in this group and it was easy for doctors to diagnose sepsis. Sep-3 is a newly proposed diagnostic criterion that has not yet been used in the studies regarding Pre; thus, we analyzed this criterion. Because the three diagnostic criteria for sepsis are significantly different, the diagnostic accuracy of Pre and the levels of Pre in the groups with and without sepsis may also differ. Of these three diagnostic criteria, only Sep-3 showed an AUC of ≥0.7 in both the <60 and ≥60 groups. Sep-3 is the most objective and reliable of these diagnostic criteria, and thus it is significant that the predictive ability of Pre for diagnosis by Sep-3 was maintained regardless of the degree of renal dysfunction.

A previous study showed that the ability of Pre to diagnose sepsis did not depend on the presence or absence of renal dysfunction or the degree of renal disorder [14–16]. Nakamura et al. investigated the ability of Pre to diagnose sepsis in ICU patients by RIFLE (Risk, Injury, Failure, Loss, End-stage kidney disease) classification and found similar AUCs in the non-acute kidney injury (AKI) group (0.784) and Risk, Injury, and Failure groups (0.698) [15]. In addition, Takahashi et al. compared the diagnostic ability of infectious diseases in 91 patients with or without AKI who met at least one SIRS criterion, and found no difference in the AUCs between the groups (AKI: 0.84, non-AKI: 0.79) [16]. These results suggest that Pre can assist

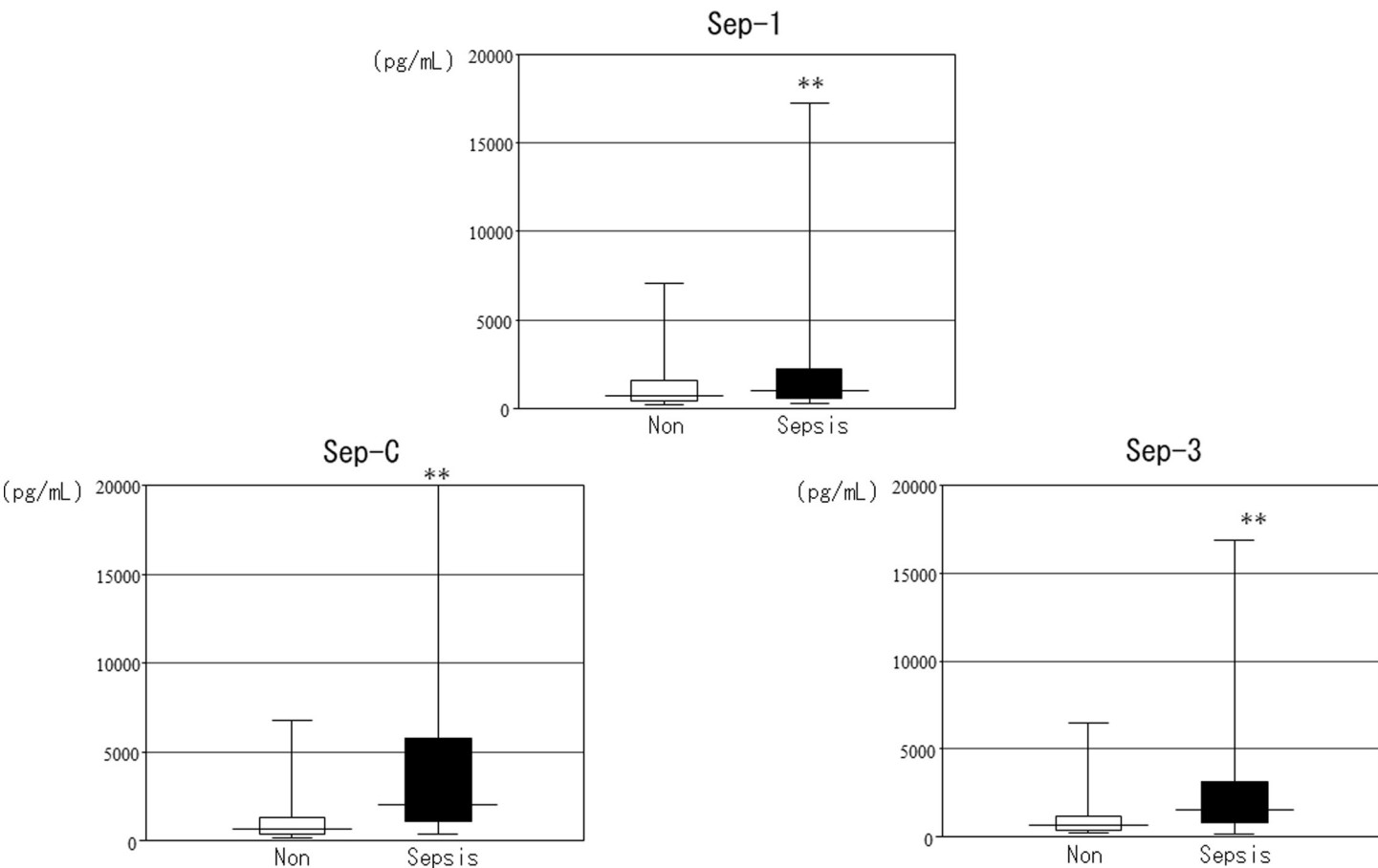

**Fig 6. Comparison of Pre levels in patients with and without each sepsis diagnosis[patients with eGFR<60 (N = 280)].** The horizontal bars represent the median, the boxes represent the IQR, and the vertical bars represent the upper and lower limits. Non: The group of non-septic patients. Sepsis: The group of septic patients. The median Pre (IQR) values are as follows: Sep-1: Sepsis (N = 138), 1014 (563–2209) vs. Non (N = 142), 692 (379–1566) pg/mL; Sep-C: Sepsis (N = 54), 2056 (1068–5775) vs. Non (N = 226), 698 (399–1284) pg/mL; Sep-3: Sepsis (N = 72), 1546 (788–3164) vs. Non (N = 157), 674 (398–1207) pg/mL. *p<0.05, **p<0.01 by the Mann–Whitney U test vs. non-sepsis.

the diagnosis of sepsis, even in patients with advanced renal dysfunction. However, it is clear that the value of Pre increases because of decreased renal function. Therefore, different cut-off values should be applied in patients with renal dysfunction.

On the basis of the Pre values in each group with and without sepsis by Sep-3 and the cut-off values obtained from ROC analysis for Sep-3, we assumed an approximate optimal Pre cut-off value for the diagnosis of sepsis in patients with renal impairment of 500 pg/mL for ≥60 and 1000 pg/mL for <60. A cut-off of 600–650 pg/mL maximized the Youden index in a previous meta-analysis [17], and 500 pg/mL has been proposed as a cut-off in Japan. However, no previous reports have determined the optimal cut-off values according to the degree of renal dysfunction. The specificity is expected to increase by setting a cut-off in persons with advanced renal dysfunction, which may help to prevent unnecessary tests and treatments. However, prospective multicenter studies are needed to verify the current proposed cut-off value. Moreover, few studies have examined the diagnostic accuracy or cut-off of Pre for the diagnosis of sepsis by Sep-3. Sep-3 has been proposed as a new diagnostic criterion for sepsis, and thus future studies on the diagnostic accuracy of Pre based on the Sep-3 criterion are needed. Considering the relationship between Pre and renal function, a recent study reported that the Pre/serum Cr ratio did not depend on the GFR stage, and the reference ratio was 67–

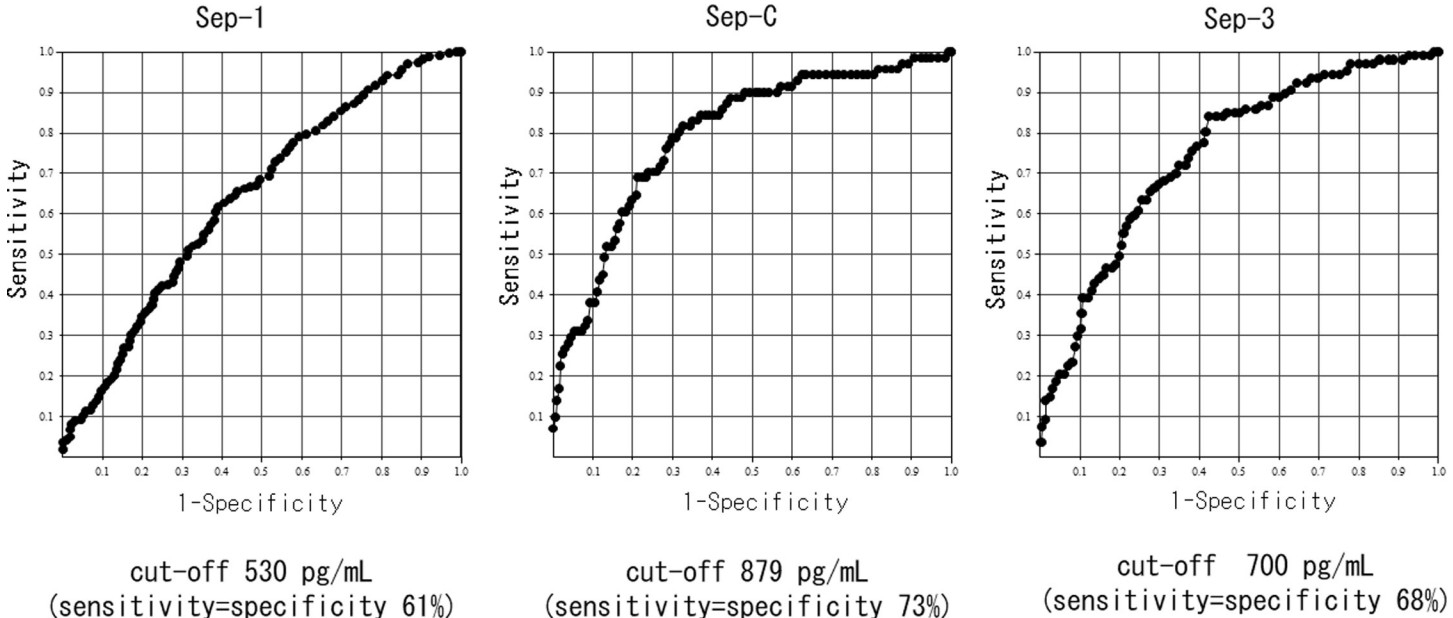

**Fig 7. Receiver operator characteristic curve of Pre for each sepsis diagnosis[All subjects with infectious diseases(N = 595)].** AUC: area under the curve. AUCs were as follows:.Sep-1: AUC = 0.64±0.02; Sep-C: AUC = 0.80±0.03; Sep-3: AUC = 0.75±0.03.

263, which was significantly higher in patients with sepsis [18]. More studies are needed to investigate whether the Pre/Cr ratio or a different cut-off setting for patients with advanced renal impairment is more useful for the diagnosis of sepsis in patients with renal dysfunction.

Several meta-analyses have examined the diagnostic accuracy of Pre for sepsis [17, 19]. Wu et al. analyzed 18 studies involving 3470 patients and reported that the pooled AUC was 0.88

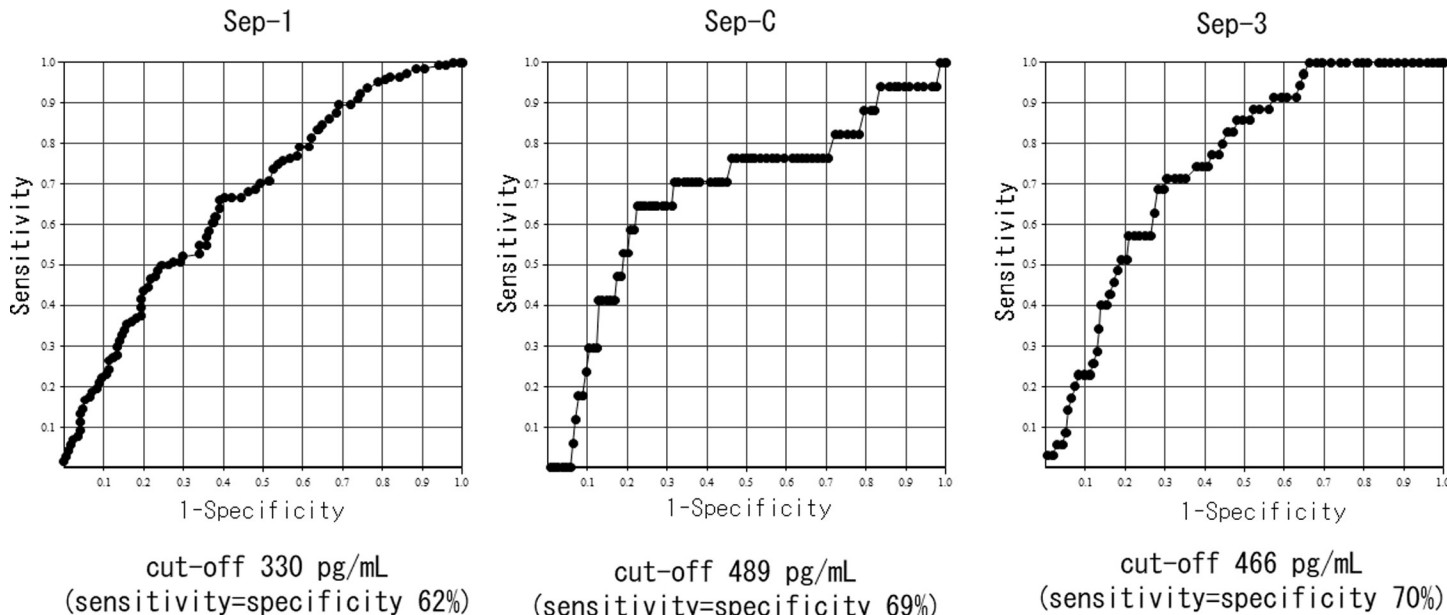

**Fig 8. Receiver operator characteristic curve of Pre for each sepsis diagnosis[patients with eGFR≥60 (N = 315)].** AUC: area under the curve. AUCs were as follows: Sep-1: AUC = 0.67±0.03; Sep-C: AUC = 0.68±0.07; Sep-3: AUC = 0.75±0.04.

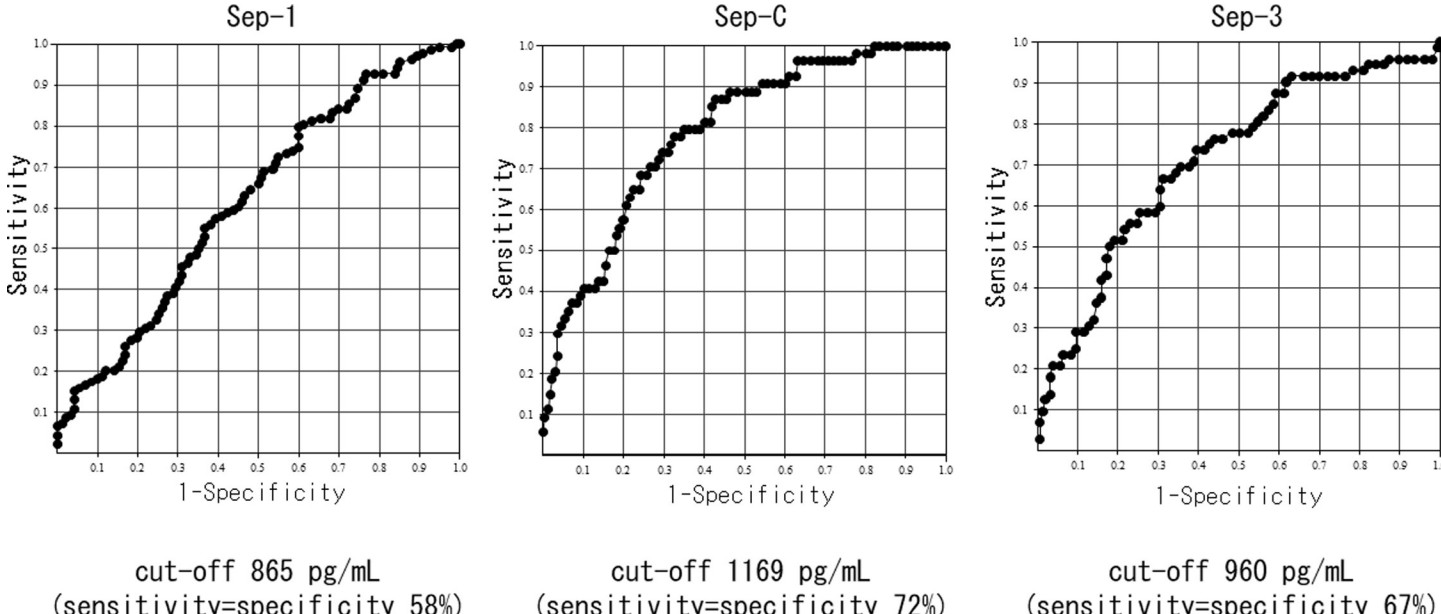

**Fig 9. Receiver operator characteristic curve of Pre for each sepsis diagnosis[patients with eGFR<60 (N = 280)].** AUC: area under the curve. AUCs were as follows: Sep-1: AUC = 0.62±0.03; Sep-C: AUC = 0.79±0.03; Sep-3: AUC = 0.71±0.04.

(95% confidence interval 0.85–0.90) [17], while another meta-analysis of 11 studies showed an area under the summary ROC curve of 0.88 (95% confidence interval 0.84–0.90) [19]. In the present study, the AUCs of Pre were 0.75 for Sep-3 and 0.80 for Sep-C, which were slightly lower than previously reported. However, meta-analyses include patients from various sources, and different controls and diagnostic criteria for sepsis [17, 19], and Wu et al. reported that the AUC was high when the controls were healthy [17]. We conducted a ROC analysis of 834 subjects, including individuals without infections, and showed that the AUC tended to be slightly higher for all diagnostic criteria. We therefore speculated that the AUC may have decreased because we evaluated the accuracy for detecting patients with sepsis among patients with infections in this study. In addition, the apparent discrepancies may be related to the diverse backgrounds of the subjects in the current study (source of patients, type of illness, severity). Conversely, the previous reports on Pre mainly diagnosed sepsis based on Sep-1, while our result was not consistent with the lowest AUC for the diagnosis by Sep-1. The cause of this discrepancy is not clear.

This study had several limitations. First, this is a retrospective single-center study that targeted only the Japanese population. Second, we used eGFR as an index of renal function, and thus it is possible that AKI cases associated with sepsis were not excluded. Additionally, the time course of sepsis was unknown, which prevented an accurate evaluation of the GFR. Third, there is a limit to the accuracy of diagnosis with infectious diseases. Fourth, approximately 14% of the patients could not be diagnosed with Sep-3. Fifth, Pre levels were expected to be related to the medical record diagnosis (Sep-C). Finally, our method of measuring serum Pre differed from previous reports.

## Conclusions

Our findings indicate that Pre is useful for diagnosing sepsis regardless of the degree of renal dysfunction by using different cut-offs. We assumed that the optimal cut-offs for the patients

in this study were 500 pg/mL for ≥60 and 1000 pg/mL for <60. Future prospective diagnostic studies on Sep-3 are needed to determine the cut-offs for patients with renal dysfunction.

## Supporting information

**S1 File. This study's dataset.**
(XLSX)

## Acknowledgments

We thank Susan Furness, PhD and Eva Lasic, PhD from Edanz (https://jp.edanz.com/ac) for editing a draft of this manuscript.

## Author Contributions

**Conceptualization:** Kimika Arakawa.

**Data curation:** Kimika Arakawa, Ayako Saeki, Reo Ide.

**Formal analysis:** Kimika Arakawa.

**Investigation:** Kimika Arakawa.

**Project administration:** Kimika Arakawa.

**Supervision:** Yoshiteru Matsushita.

**Writing – original draft:** Kimika Arakawa.

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
