## [Decision Letter · Decision Letter 0]

6 Jul 2022

PONE-D-22-06047Presepsin cutoff value for diagnosis of sepsis in patients with renal dysfunctionPLOS ONE

Dear Dr. Arakawa, 

Thank you for submitting your manuscript to PLOS ONE. After careful consideration, we feel that it has merit but does not fully meet PLOS ONE’s publication criteria as it currently stands. Therefore, we invite you to submit a revised version of the manuscript that addresses the points raised during the review process.

We look forward to receiving your revised manuscript.

Kind regards,

Donovan Anthony McGrowder, PhD., MA., MSc

Academic Editor

PLOS ONE

Journal Requirements:

Additional Editor Comments:

Dear Dr. Arakawa,

The manuscript was revised in accordance with the reviewers’ comments and is provisionally accepted pending final checks for formatting and technical requirements.

Regards,

Dr. Donovan McGrowder (Academic Editor)<o:p></o:p>

Reviewers' comments:

Reviewer's Responses to Questions

**Comments to the Author**

1. Is the manuscript technically sound, and do the data support the conclusions?

Reviewer #1: Yes

Reviewer #2: Partly

Reviewer #3: Partly

2. Has the statistical analysis been performed appropriately and rigorously? 

Reviewer #1: Yes

Reviewer #2: No

Reviewer #3: No

3. Have the authors made all data underlying the findings in their manuscript fully available?

Reviewer #1: Yes

Reviewer #2: Yes

Reviewer #3: Yes

4. Is the manuscript presented in an intelligible fashion and written in standard English?

Reviewer #1: Yes

Reviewer #2: Yes

Reviewer #3: No

5. Review Comments to the Author

Reviewer #1: Kimila Arakawa et al. investigated the effects of eGFR on the diagnostic accuracy of presepsin and determine the optimal cutoff value in the patients with renal impairment. Log-transformed presepsin showed a significant negative correlation with eGFR. In the ROC analysis, AUC for sepsis-1 was the lowest, and Sep-C and sepsis-3 were moderately accurate. The authors suggested the optimal cutoff 500 pg/mL for ≤G3a and 1500 pg/mL for ≥G3b. Although the aim of the study is clear and straightforward, the advantage of measuring presepsin in the clinical setting is not established.

Major comments

1. In lines 45-47, it is not clear how were the enrolled patients selected to measure presepsin levels. Were they suspected to have sepsis or infection? Were they positive for some of the diagnostic criteria of the sepsis-1, sepsis-3 or Sep-C? The inclusion criteria were not clear.

2. The authors employed sepsis-1, sepsis-3 or Sep-C as gold standards, presepsin will not exceed the diagnostic values of these diagnostic criteria. The authors should set more valuable clinical outcomes for gold standard, e.g. prognosis of the patients. In addition, the sepsis-1, sepsis-3 or Sep-C were defined for the quick diagnosis for sepsis, the measurement of presepsin may not contribute the fast and easy diagnosis of sepsis.

3. In line 110-111, why were SOFA and qSOFA not available? Were the data missing? How did the authors analyze the missing data?

4. In Fig 1, eGFR of some of the patients was extremely high ranging from 150-400 mL/min.

5. In Fig 5, cut-off values are indicated in the legends, they should be indicated in the figure.

6. The authors should make clear the benefit of the measurement of presepsin in the clinical setting.

Reviewer #2: Major points

The authors investigated the cutoff value of presepsin for detecting sepsis in patients with renal insufficiency. Although the results were meaningful, there were several problems. Especially, I think that it is difficult to determine the cutoff values of presepsin levels from this study design because of single center study. However, they described that 500 and 1500 pg/ml has been proposed as a cutoff in Japan (Line 240-241). Additionally, it is difficult using this cutoff value in clinical setting. If there are two infectious patients with eGFR of 40 and 50, the cutoff value will be 1500 and 500, respectively. Please show the reasons why the assessments of presepsin levels were different between definition of sepsis, Sep-1, Sep-C and Sep-3, in the Discussion.

Minor points

Please describe the detail of the definition of Sep-C.

The description of G1, G2, G3a, G3b, G4 and G5 are inappropriate, because these are used for CKD patients. Probably, there were many patients with AKI or acute exacerbation of CKD. I recommend using ≥90, ≥60 to <90, ≥45 to <60, ≥30 to <45, ≥15 to <30, <15, and dialyzed patients.

Please describe presepsin levels as median and range or median and interquartile range (IQR). The correlation between presepsin and eGFR should be assessed by non-parametric Spearman's correlation test for two reasons. First, it is unclear whether log presepsin was normally distributed. Second, eGFR seems to be not-normally distributed because eGFR at baseline was reported to be 65.1 ± 37.0.

Table 1 should be presented both whole patients and by presepsin levels, the exposure of interest. It will make easier to understand the clinical characteristics of the patients with higher (or lower) presepsin levels.

Reviewer #3: Arakawa et al described cutoff value for diagnosis of sepsis in patients with renal impairment. This is a potentially important manuscript to provide a clinical important indicator to diagnosis of sepsis in patients with reduced GFR. However, the manuscript has several flaws which need to be addressed.

Major criticism

1. eGFR was calculated from serum creatinine in patients with acute kidney injury (AKI) or with acute kidney disease (AKD). Serum creatinine is not consistent in the condition of AKI and AKD. In addition, the increasing serum creatinine is delayed from falling true GFR in developing to AKI, while this could be happen vice versa in recovering from AKI. Thus, in the beginning of sepsis, the calculated eGFR could be often higher than true measured GFR, leading to overestimation of true GFR and in the recovering from sepsis, eGFR could be lower than measured GFR. The authors should mention this point in limitation.

2. In Fig. 1, some participants were overestimated GFR whose eGFR>200mL/min/1.73m2. I feel these participants might lose muscle mass. I suggest these participants should be excluded from this study.

3. In Fig. 3, approximately 50% of participants were not able to diagnose sepsis or not in each CKD category because of lack of data. I think Sep-3 criteria should not be used for diagnosis of sepsis in this study.

4. The authors concluded that the optimal cutoff values for presepsin in patients with renal impairment should be approximately 500 pg/ml for ⪯G3a and 1500 pg/mL for ⪰G3b. I think this conclusion solely comes from the results of Sep-C. In addition, these cutoff values might be for only Japanese because this study was conducted only Japanese patients.

Minor criticism

1. The manuscript contains Figure legends in the main text. Figure legends should be separately noted.

2. The authors showed LogPre differed significantly between patients with and without sepsis. I think the authors also should show the serum presepsin values differ between patients with and without sepsis.

---

## [Author Response · Author response to Decision Letter 0]

16 Aug 2022

We thank the reviewers and editor's comments. We revised our manuscript to response all comments.

---

## [Editor Report · Decision Letter 1]

18 Aug 2022

Presepsin cut-off value for diagnosis of sepsis in patients with renal dysfunction

PONE-D-22-06047R1

Dear Dr. Arakawa, 

We’re pleased to inform you that your manuscript has been judged scientifically suitable for publication and will be formally accepted for publication once it meets all outstanding technical requirements.

Kind regards,

Donovan Anthony McGrowder, PhD., MA., MSc

Academic Editor

PLOS ONE

Additional Editor Comments:

Dear Dr. Arakawa,

The manuscript was revised in accordance with the reviewers’ comments and is provisionally accepted pending final checks for formatting and technical requirements.

Regards,

Dr. Donovan McGrowder (Academic Editor)<o:p></o:p>

---

## [Editor Report · Acceptance letter]

23 Aug 2022

PONE-D-22-06047R1 

Presepsin cut-off value for diagnosis of sepsis in patients with renal dysfunction 

Dear Dr. Arakawa:

I'm pleased to inform you that your manuscript has been deemed suitable for publication in PLOS ONE. Congratulations! Your manuscript is now with our production department. 

Kind regards, 

on behalf of

Dr. Donovan Anthony McGrowder 

Academic Editor

PLOS ONE